# Z-score Normalized SAC Plus Behavioural Cloning for Offline Reinforcement Learning

## Abstract

Reinforcement learning (RL) defines the task that optimize a policy to maximize the cumulative reward function. Online learning collects data samples by interacting with the environment of task. Instead, Offline RL learns effective policies from a prior demonstrated dataset, which has the potential to transfer the successes between tasks. The main challenge encountered by offline RL is the inaccurate value estimates from out-of-distribution (OOD) actions, and applying vanilla off-policy algorithms to offline setting will cause severe overestimation bias for actions beyond the dataset distribution, because of the disability to correct value estimation errors via observations from the environment. To tackle this problem, the behavior regularization has been adopted in the literature to prevent the selected actions far away from the distribution of dataset so that the learned policy can be optimized within the support set of dataset. One simple method is combining RL with the behavioural cloning (BC) linearly. By making a right balance of the relative weight between RL and BC, the pre-existing off-policy algorithms are able to work efficiently offline at the minimal cost of complexity. Overly large BC term will limit the agents potential to explore better policy, and oversize RL term will cause more OOD actions, both of which are undesired. Simulated by TD3-BC, this paper aim to make a more efficient offline RL algorithm at the cost of minimal changes and light complexity. We find that the BC term can be added to the policy update of SAC algorithm to get extensively better performance with proper weight adjustment and gradient self-adaption. The proposed SAC-BC algorithm is evaluated on the D4RL benchmark and proved to converge to much higher levels due to better exploration provided by tuned maximum entropy.

## 1 Introduction

Offline reinforcement learning (RL) aims to find the optimal policy to maximize the cumulative reward based on a fixed collected dataset. Despite the extra difficulty to learn a good policy, offline RL removes the necessity to collect expensive and risky data from the environment in real-world applications. However, offline RL is more challenging than traditional online RL since the agent cannot receive observations from the environment for policy exploration and correcting the extrapolation error, which is defined as the poor value estimates of state-action pairs outside the given dataset. This bootstrapping errors will accumulate much faster during training when using the fixed demonstration dataset than using the online data, causing highly sub-optimal policies or even complete divergence Levine et al. (2020).

The general approach to tackling the extrapolation problem is a family of behavior regularization, which constrain the policy update within the nearby region of dataset distribution. The nearby region is usually measured by the divergence between the learnt policy and the behavior policy, which is computed by some kernel approaches like Kernel Maximum Mean Discrepancy (MMD) Gretton et al. (2007), Wasserstein distance and KL divergence Wu et al. (2019b). One simplistic and computationally efficient way is to incorporate behavioural cloning (BC) into existing off-policy algorithms. Such kind of method is able to seek better policies than more complex alternatives Fujimoto & Gu (2021) at the cost of minimal changes, which produce less requirement of redundant hyperparameters. The minor modifications make the underlying algorithms they are based on easier to generalize to better performance.

The role of BC is to limit the policy exploration induced by the underlying off-policy algorithms, so there exists a tradeoff between policy exploration and policy regularization. It is significant to determine the optimal contribution of BC term. Specifically, too small weight of BC term will smooth the propagation of extrapolation errors by driving actions away from the dataset, and too large BC contribution will limit the agents potential to explore better policy. Within the scope of literature, the level of BC is maintained as constant to keep a stable performance. Besides adjusting the BC level, finding a suitable policy exploration method is another key to better policies.

Many proposed offline approaches make nontrivial adjustments while claiming to be simple. Specifically, they may include additional hyperparameters that are hard to tune or secondary components like generative models, which make the reproducibility of these offline algorithms difficult Henderson et al. (2018); Tucker et al. (2018); Engstrom et al. (2019); Andrychowicz et al. (2020); Furuta et al. (2021). Minimalist approaches have modest computation complexity and are easy to transfer techniques across algorithms. In this work we try to seek other underlying off-policy algorithms that can work well with BC. Accordingly, the influence of BC term should be adjusted to fit the foundation algorithms. Inspired by TD3-BC Fujimoto & Gu (2021) which adding BC to the twin delayed deep deterministic policy gradient (TD3) algorithm Fujimoto et al. (2018), we build our offline algorithm on the basis of soft actor-critic (SAC) Haarnoja et al. (2018), which is mentioned by Nair et al. (2020) as a baseline to accelerating online learning with the help of offline dataset, but has not yet been well studied to apply to offline setting. One possible reason for this absence of research may be because the maximum entropy learning tends to encourage the policy to explore actions outside the nearby region of fixed dataset, and thus makes it difficult to stabilize the offline learning for long.

In this work, we first reorganize the combination of SAC and BC algorithm on the basis of TD3-BC. Furthermore, we adopt the state normalization from TD3-BC and the modified Q-value from SAC as the foundation. The modified Q-value is the combination of Q-value with a entropy term, and it is applied to both the policy evaluation and policy update. In the policy update step, we perform the z-score normalization of the modified Q-value to drive it close to the standard normal distribution. Then the z-score normalized modified Q-value can be balanced to adapt to the value of BC term, and the the gradient of policy objective can be self-adapted to mitigate the additional extrapolation error induced by the entropy exploration. By this means, the gradient descent of the policy update will be reduced when the given dataset is high, so that the dditional extrapolation error induced by the entropy exploration can be mitigated. Our proposed approach is evaluated on the D4RL benchmark of MuJoCo tasks to see its benefits.

## 2 RELATED WORK

Except for TD3-BC which directly incorporates BC into the policy updates, there are other methods using BC in different ways. For example, the implicit Q-learning (IQL) Kostrikov et al. (2021a) learns the value function with expectile regression and combine it with advantage-weighted behavioral cloning to update policies without explicitly modeling the dataset distribution based on the in-distribution actions. Fisher-BRC Kostrikov et al. (2021a) regularize the critic update using the gradient penalty via the Fisher divergence metric. Both methods have comparable performance with TD3-BC but lose advantage in complexity, especially when the dataset of offline learning is less expert Beeson & Montana (2022).

Some methods apply other forms of behavior regularization. For example, the authors in Fujimoto et al. (2019); Wu et al. (2019a) train a Conditional Variational AutoEncoder (CVAE) Sohn et al. (2015) to model the distribution of dataset, and Wu et al. (2019a); Kumar et al. (2019) utilizes some kind of divergence metrics to regularize the policy updating process. Although these methods try to combat the negative impact of extrapolation error, they seem to have worse performance than TD3-BC, especially when the fixed dataset is less expert and hard to be modeled accurately. The implicit cause may lie in the unbalanced tradeoff between reducing the extrapolation error and utilizing policy exploration.

Imitation learning has also been combined with RL to make use of expert demonstrations, including mixing with adversarial methods Zhu et al. (2018); Kang et al. (2018), using the offline dataset for initial training Pfeiffer et al. (2018), making modifications in the replay buffer to store online data Vecerik et al. (2017); Paine et al. (2019), making proper designs for the action-value function Kim

et al. (2013); Hester et al. (2018) and improving the method of reward shaping Judah et al. (2014); Wu et al. (2021).

Throughout the literature of offline RL, the provided demonstrations from the given fixed dataset are either used to improve the learning speed of online learning or adopted to tackle the tradeoff between exploration and extrapolation inside the offline learning. This paper focuses on the offline learning itself and attempt to explore a good policy while keeping the agent staying close to the offline dataset. Previous literature built a parameterized model to fit the behavior policy to estimate the action distribution of dataset. Some works directly use the model to parameterize the leant policy Fujimoto et al. (2019); Ghasemipour et al. (2021), and some others utilize the model of behavior policy to define some kind of divergence metrics for regularization Jaques et al. (2019); Kumar et al. (2019); Wu et al. (2019b); Siegel et al. (2020); Guo et al. (2020); Kostrikov et al. (2021a). Instead, in this paper we directly perform BC by the expectation form of mean square error (MSE) between the actions from the learnt policy and the actions from a truncated subset of the dataset.

## 3 BACKGROUND

### 3.1 ONLINE REINFORCEMENT LEARNING

We inherit the definition of previous works and set the RL problem based on Markov Decision Process (MDP), which is simplified as the tuple $(\mathcal{S}, \mathcal{A}, T, R)$, consisting of the state space $\mathcal{S}$, the action space $\mathcal{A}$, the transition distribution $T(\cdot|s, a)$ and the reward function $R(s, a)$, where $s \sim \mathcal{S}$ and $a \sim \mathcal{A}$ represent the observation and the action from learnt or behavior policy, respectively. In online learning, the behavior policy is followed by the agent to interact with the MDP environment to collect observations $(s, a, r, s')$, which will be stored in a replay buffer to get the online dataset. Among the observation slot, the next state $s'$ is observed from the environment and implicitly determined by the the distribution of transition dynamics $T(s'|s, a)$, and the immediate reward $r$ follows the reward function $R(s, a)$. The set of policy $\pi$ can be either deterministic or stochastic. For stochastic policies, the Gaussian distribution is usually selected to stimulate generalization and simplicity. Following the transition distribution and policy distribution, the agent produces a series of the observation slots During a rollout of timesteps. A discount factor $\gamma \in [0, 1]$ is adopted to weight the contribution of each observation, and the goal of RL is to find an optimal policy to maximize the expectation of discounted cumulative reward (expected return), which is formulated as $\mathbb{E}_\pi \left[ \sum_{t=0}^{\infty} \gamma^t r(s_t, a_t) \right]$.

When the actor-critic method is applied to RL, the training will alternate between the critic update (policy evaluation) step and the actor update (policy improvement) step. The large-scale continuous control problems are usually addressed by deep RL, which adopt function approximation to parameterize the Q-value function as $Q_\pi(s, a)$, which estimates the expected return of a rollout starting from the state-action pair $(s, a)$ and following the policy $\pi$, where the actor as modeled as $a = \pi(s)$ for deterministic policy and $\pi(a|s)$ for stochastic policy, respectively. The policy evaluation is normally performed based on the Bellman equation $Q_\pi(s, a) = r(s, a) + \mathbb{E}_{s' \sim T, a' \sim \pi} [Q_\pi(s', a')]$ if no value penalty is given, and the policy improvement maximizes $Q_\pi(s, a)$ to update the actor parameters without policy regularization.

### 3.2 OFFLINE REINFORCEMENT LEARNING

When it comes to the offline learning, a pre-collected dataset is adopted to replace the observations from the interactions with the environment. If the given dataset provides expert demonstrations, the agent can easily achieve optimal policies on the basis of BC. Otherwise, offline RL struggle to maintain stable performance when the dataset only gives sub-optimal demonstrations due to the extrapolation error, which tends to propagate and accumulate during training and causes instability or even complete failure Levine et al. (2020). Therefore, the ability to learn effective policies from non-optimal demonstrations becomes a significant metrics to measure the quality of offline RL algorithms.

As the most related work to this paper, TD3-BC is simply adapted from TD3 to apply to the offline setting by adding a weighted BC term to the policy updates, where the BC term pushes the agent to stay close to demonstrations from the given dataset $\mathcal{D}$. According to Fujimoto & Gu (2021), the

policy update of TD3-BC is illustrated as

$$\pi = \arg\max_{\pi} \mathbb{E}_{(s,a)\sim\mathcal{D}} \left[ \lambda Q(s, \pi(\cdot|s)) - (\pi(\cdot|s) - a)^2 \right], \tag{1}$$

where the hyperparameter $\lambda$ tunes the relative contribution of BC term, because the BC term is upper-bounded by $4$ if the actions range within $[-1, 1]$ while the RL term is highly susceptible to the estimated Q-value. To make the RL term comparable to the BC term, $\lambda$ is chosen to put the RL term in proportion to the normalized Q-value, which is given by

$$\lambda = \frac{\rho}{\mathbb{E}_{(s,a)\sim\mathcal{D}_i} |Q(s,a)|}, \tag{2}$$

where $\rho$ is a constant hyperparameter, and $\mathcal{D}_i$ represents the sampled mini-batch of the $i$-th time step from the dataset. (2) means that the Q-value is normalized by the mean absolute values of Q-values.

The other trick utilized by TD3-BC is normalizing the states to have $0$ mean and unit variance. Let $s_i$, $\mu_i$ and $\sigma_i$ be the sampled states and their mean and standard deviation, then the normalized states are computed by

$$s_i \leftarrow \frac{s_i - \mu_i}{\sigma_i + \epsilon}, \tag{3}$$

where $\epsilon$ is the offset constant.

## 4 REINFORCEMENT LEARNING PLUS BEHAVIOURAL CLONING

### 4.1 CHALLENGES FACED BY RL-BC

The loose computation complexity and advantageous performance of the linear combination between RL and BC makes it preferable to be applied in offline RL. Although RL-BC has comparable performance with some other state-of-the-art offline RL algorithms with little cost, however, its efficiency is challenged by tuning the hyperparameter that weights the relative importance of the RL term to the BC term. We use $\theta$ to reparameterize the policy as $a \sim \pi_\theta(\cdot|s)$ and use $J(\theta)$ to represent the policy objective of (9). Then the gradient of $J(\theta)$ with respect to $\theta$ can be written as

$$\hat{\triangledown}_\theta J(\theta) = \mathbb{E}_{(s,a)\sim\mathcal{D}_i} \left[ \hat{\triangledown}_{\hat{a}} Q(s, \hat{a}) \hat{\triangledown}_\theta \hat{a} \left[ \lambda - 2(\hat{a} - a) \right] \right], \tag{4}$$

where $\hat{a} \sim \pi_\theta(\cdot|s)$ for the stochastic policy and $\hat{a} = \pi_\theta(s)$ for the deterministic policy.

During the training process, we expect the policy to be optimized to increase the expected return, i.e., $\hat{\triangledown}_{\hat{a}} Q(s, \hat{a}) \geq 0$. Then if the sign of $\hat{\triangledown}_\theta \hat{a}$ is determined to be unchanged, the monotonicity of $J(\theta)$ is highly limited by the selected value of $\lambda$. However, the sign of $\hat{\triangledown}_{\hat{a}} Q(s, \hat{a})$ will be changed multiple times throughout training. It should be noted that $\hat{a} - a$ ranges from $-2$ to $2$ if the actions range within $[-1, 1]$. If the value of $\lambda$ is lower-bounded by $2$, the sign of $\hat{\triangledown}_\theta J(\theta)$ will be the same as $\hat{\triangledown}_{\hat{a}} Q(s, \hat{a})$. Otherwise, the value of $\hat{\triangledown}_\theta J(\theta)$ is not closely related to $\hat{\triangledown}_{\hat{a}} Q(s, \hat{a})$. This is to say, even when the update of (4) meets the fixed point, i.e., $\hat{a} - a = \lambda/2$, $\hat{\triangledown}_{\hat{a}} Q(s, \hat{a})$ can still be positive. Furthermore, since $\hat{a} - a = \lambda/2$ is the fixed point condition, the value of $\lambda$ determines the nearby region of mini-batches from the fixed dataset. Small value of $\lambda$ will ensure the stability of training but does not necessarily limit the final performance as long as the Q-value function keeps considerable gradient with respect to the action.

It is worthy to improve the gradient of expected return with respect to the action while keeping a small value of $\lambda$. However, $\hat{\triangledown}_{\hat{a}} Q(s, \hat{a})$ is uncontrollable by performing the policy update of (9). From numerous experimental results, we also find that there exists a lower-bound for $\lambda$ to make a fine asymptotical performance. In other words, the RL-BC method meets a challenge when choosing a proper value of $\lambda$ for updating (9).

## 5 VARIANT OF SAC PLUS BEHAVIOURAL CLONING

The most significant problem of offline RL is the extrapolation error, which can be interpreted as the difficulty to properly identity and correct OOD actions. There have been several approaches to this problem, including imitating, constraining, regularizing the policy so that the learnt policy will

| | | Expert | Medium Expert | Medium Replay | Medium | random |
|---|---|---|---|---|---|---|
| **TD3-BC** | Ant | 117 | 112 | 95 | 108 | 32 |
| | HalfCheetah | 95 | 45 | 45 | 47 | 12 |
| | Hopper | 110 | 60 | 25 | 58 | 8 |
| | Walker2d | 110 | 80 | 72 | 82 | 4 |
| **SAC-BC** | Ant | 53 | 97 | 82 | 110 | 40 |
| | HalfCheetah | 95 | 90 | 45 | 50 | 15 |
| | Hopper | 108 | 98 | 38 | 48 | 33 |
| | Walker2d | 110 | 110 | 85 | 82 | 2 |

Table 1: **Average normalized score over the final 10 evaluations**

not be overly contaminated by the OOD actions beyond the dataset. Among the existing divergence metrics for policy regularization, the KL divergence seems to be commonly adopted in the literature. However, there is no convinced theoretical analyses to argue the conspicuous advantage of one divergence or distance metric over others. Therefore, instead of basing an algorithm on complicated logics and computation, we try to maintain minimal but delicate modifications to some computation-efficient approaches, like the pre-existing SAC. According to Fujimoto & Gu (2021), such kind of minimalist approach has several merits, since it increases the scalability and generalization for analysis by constraining the number of hyperparameters that need to be tuned and reducing the computation complexity.

The most related work to our paper is TD3-BC, which combines a normalized TD3 term with a weighted BC term for policy improvement. Besides, another potential similar approach is the "SAC+BC" that is used as a baseline in Nair et al. (2020). However, it is highly possible that this "SAC+BC" is built in the same way with TD3-BC because it performs badly in Nair et al. (2020). As we have observed in our experiments, even combining SAC with BC based on the changes made by TD3-BC, the hyperparameter should be retuned to balance the extra policy exploration brought by the maximum entropy. We adopt the minimal modifications proposed in Fujimoto & Gu (2021) to SAC and organize the new SAC-BC as

$$\pi = \arg\max_{\pi} \mathbb{E}_{(s,a)\sim\mathcal{D}} \left[ \lambda Q^{\alpha}(s, \pi(\cdot|s)) - (\pi(\cdot|s) - a)^2 \right], \tag{5}$$

where $Q^{\alpha}(s, \pi(\cdot|s))$ represents the Q-value function minus the entropy term, given by

$$Q^{\alpha}(s, \pi(\cdot|s)) = Q(s, \pi(\cdot|s)) - \alpha \log \pi(\cdot|s). \tag{6}$$

Accordingly, the normalized term $\lambda$ is chosen to put the RL term in proportion to the normalized combination of Q-value and entropy,, which is given by

$$\lambda = \frac{\rho}{\mathbb{E}_{(s,a)\sim\mathcal{D}_i} |Q^{\alpha}(s, a)|}, \tag{7}$$

where $\rho$ is a constant hyperparameter, and $\mathcal{D}_i$ represents the sampled mini-batch of the $i$-th time step from the dataset. Applying the according setting of hyperparameter as $\rho = 2.5$ to the SAC-BC, and the mean of final performance of SAC-BC is reported in Table 1.

## 6 Z-SCORE NORMALIZED SAC PLUS BEHAVIOURAL CLONING

In this part, we describe our new offline algorithm that is improved from the recreated SAC-BC proposed in (5). The process follows several steps. Firstly, we normalize the features of every state in the fixed dataset as (3). Secondly, we compute the Q-value function minus the entropy term according to (6). Thirdly, we adopt the z-score normalization to address the features of the modified Q-value at the learnt policy and every state in the given dataset, which is shown in (8).

$$Q^{\alpha}(s_i, \pi(\cdot|s_i)) \leftarrow \frac{Q^{\alpha}(s_i, \pi(\cdot|s_i)) - \mu_{Q^{\alpha}}}{\sigma_{Q^{\alpha}(s_i, \pi(\cdot|s_i))}}, \tag{8}$$

where $s_i$ is the $i$-th feature of $s$, $\mu_{Q^\alpha}$ and $\sigma_{Q^\alpha(s_i, \pi(\cdot|s_i))}$ represent the mean and standard deviation of the modified Q-value across all features. Different from $\mu_{Q^\alpha}$ which is constant at every policy update, $\sigma_{Q^\alpha(s_i, \pi(\cdot|s_i))}$ is connected with the trainable policy parameter. The last step but not least, the policy update of z-score normalized SAC-BC (ZNSAC-BC) is shown as

$$\pi = \arg\max_\pi \mathbb{E}_{(s,a)\sim\mathcal{D}} \left[ \rho \frac{Q^\alpha(s, \pi(\cdot|s)) - \mu_{Q^\alpha}}{\sigma_{Q^\alpha(s_i, \pi(\cdot|s_i))}} - (\pi(\cdot|s) - a)^2 \right], \tag{9}$$

where $\rho$ is a constant hyperparameter.

Although normalization has been commonly adopted in a lot of deep RL methods, it is noticeable that the z-score normalization of Q-value is novel, which is distinctly different from the state normalization in (3), and thus has additional benefits. Firstly, the distribution of z-score normalized modified Q-value is close to standard normal distribution, so that it can be balanced to suit the value of BC term. Secondly, the approximated standard normal distribution can avoid the sensitivity to the absolute value of Q-value used in (2). Most importantly, the trainable variance from $\sigma_{Q^\alpha(s_i, \pi(\cdot|s_i))}$ will dynamically balance the relative contribution between RL and BC. It will also auto-tune the gradient of policy objective, which is computed by (13), to mitigate the additional extrapolation error induced by the entropy exploration, and thus improve the robustness of training.

In the actor-critic paradigm, the policy and the modified Q-value function are usually parameterized as the actor parameter $\theta$ and the critic parameter $\omega$, which is symbolized as $\pi_\theta$ and $Q_\omega^\alpha$, respectively. Accordingly, $\theta'$ and $\omega'$ define the parameters of target actor and target critic, respectively. The loss function of the critic update can be estimated by

$$L(\omega) = \mathbb{E}_{(s,a,r,s')\sim\mathcal{D}} \left[ \frac{1}{2} (r + \gamma Q_{\omega'}^\alpha(s', a') - Q_\omega(s, a))^2 \right], \tag{10}$$

where $a' \sim \pi_{\theta'}(\cdot|s')$ for the stochastic policy and $a' = \pi_{\theta'}(s')$ for the deterministic policy, and $(s, a, r, s')$ is a tuple of mini-batch sampled from the given dataset. By minimizing (10), the critic parameter can be updated for the policy evaluation step. Then (10) can be optimized with stochastic gradient as

$$\hat{\nabla}_\omega L(\omega) = \mathbb{E}_{(s,a,r,s')\sim\mathcal{D}}[\hat{\nabla}_\omega Q_\omega(s, a)(Q_\omega(s, a) - r - \gamma Q_{\omega'}^\alpha(s', a'))]. \tag{11}$$

The surrogate objective function of updating the current actor parameter $\theta$ in the policy improvement step can be given by

$$J(\theta) = \mathbb{E}_{(s,a)\sim\mathcal{D}} \left[ \rho \frac{Q_\omega^\alpha(s, \pi_\theta(\cdot|s)) - \mu_{Q_\omega^\alpha}}{\sigma_{Q_\omega^\alpha(s_i, \pi(\cdot|s_i))}} - (\pi_\theta(\cdot|s) - a)^2 \right], \tag{12}$$

where $\mu_{Q_\omega^\alpha}$ and $\sigma_{Q_\omega^\alpha}$ represent the mean and standard deviation of the modified Q-value parameterized by $\omega$. By maximizing (12), the actor parameter can be updated for policy improvement each step. The gradient of (12) is computed as

$$\hat{\nabla}_\theta J(\theta) = \mathbb{E}_{(s,a)\sim\mathcal{D}} \left[ \frac{\rho \hat{\nabla}_\theta Q_\omega^\alpha(s, \pi_\theta(\cdot|s))}{\sigma_{Q_\omega^\alpha(s_i, \pi(\cdot|s_i))}} - 2\hat{\nabla}_\theta \pi_\theta(\cdot|s)(\pi_\theta(\cdot|s) - a) \right]. \tag{13}$$

Then the target parameters $(\omega', \theta')$ are updated following the "soft" target updates Lillicrap et al. (2015) by $(\omega, \theta)$, in the way of

$$\begin{aligned} \omega'_{t+1} &\leftarrow \tau\omega_{t+1} + (1-\tau)\omega'_t, \\ \theta'_{t+1} &\leftarrow \tau\theta_{t+1} + (1-\tau)\theta'_t, \end{aligned} \tag{14}$$

where $0 \le \tau < 1$ is the factor to control the speed of policy updates for the sake of small value error at each iteration.

Eqs. (6), (8), (10) and (12) have summarized all the modifications to SAC, and the according pseudocode is described by Algorithm 1.

## 7 EXPERIMENTS

We evaluate our ZNSAC-BC algorithm on the D4RL benchmark of MuJoCo tasks Todorov et al. (2012); Brockman et al. (2016); Fu et al. (2020), which incorporates datasets of several experimental

---

**Algorithm 1** Z-score Normalized SAC-BC Algorithm

---

1: Initialize parameters $\omega \leftarrow \omega_0$, $\theta \leftarrow \theta_0$
2: Initialize target parameters $\omega' \leftarrow \omega'_0$, $\theta' \leftarrow \theta'_0$
3: Initialize the learning rates $l_c$, $l_a$ for the critic and the actor, the time step $t \leftarrow 0$, the soft update hyperparameter $\tau$, the maximum time step $T$, the batch size $B$ and the fixed dataset $\mathcal{D}$.
4: **while** $t < T$ **do**
5:     Sample a batch of transitions $\mathcal{B} = (s, a, r, s')_{i=1}^{B}$ from $\mathcal{D}$
6:     **for** each time step **do**
7:         Normalizing the modified Q-value towards z-score following (8)
8:         $\omega_{t+1} \leftarrow \omega_t - l_c \nabla_{\omega_t} L(\omega_t)$ following (11)
9:         $\theta_{t+1} \leftarrow \theta_t + l_a \nabla_{\theta_t} J(\theta_t)$ following (13)
10:       $(\omega', \theta')_{t+1} \leftarrow \tau(\omega, \theta)_{t+1} + (1 - \tau)(\omega', \theta')_t$ following (14)
11:     **end for**
12:     $t \leftarrow t + 1$
13: **end while**

---

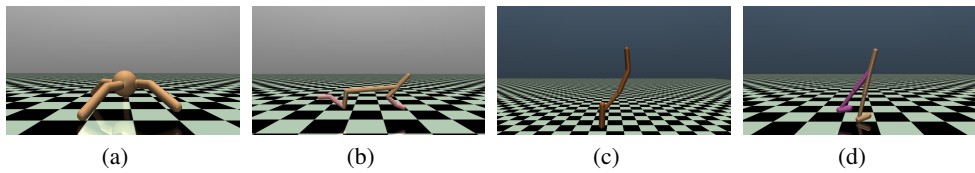

| (a) | (b) | (c) | (d) |

Figure 1: (a) Ant-v2; (b) Halfcheetah-v2; (c) Hopper-v2; (d) Walker2d-v2

settings. We select Ant, Hopper, walker2d and Halfcheetah as baselines and evaluate the proposed algorithm on their random, medium, medium-replay, medium-expert and expert datasets. The baselines we choose are several state-of-the-art offline RL algorithms including batch-constrained deep Q-learning (BCQ) Fujimoto et al. (2019), IQL Kostrikov et al. (2021b), TD3-BC Fujimoto & Gu (2021) and SAC-BC.

To keep the experimental evaluation relatively fair, we share the same set of hyperparameters for each benchmark across the proposed algorithms and other baselines. We set the maximum number of time steps as $10^6$ for every algorithm and evaluate it by running some real rollouts every 5000 time steps. Each evaluation process is composed of 10 episodes by interacting with the environment, and the mean of their results are used to stand for the average reward. We record the final performance results and show the learning curves in Fig. 2. From these figures, we can observe that the performance of ZNSAC-BC is the best across all tasks, especially when the given dataset is less expert.

## 8 CONCLUSION

In this paper, we proposed a state-dependent adaptive temperature to encourage more random actions and boost training, which can meet a better balance between the efficiency and stability. Then we add an extra term containing the asymptotic maximum entropy to stabilize the convergence. The two components are combined with the selected critic value to function as the target Q-value and the surrogate objective in the policy evaluation and improvement steps. Based on the above two components, we present AAAC, which is based on adaptive and asymptotic maximum entropy combined with actor-critic random policies, to form strong adaptation to the tradeoff between efficiency and stability, and provide more exploration and flexibility to avoid the problem of saddle point. We evaluate our method on a set of Gym tasks, and the results show that the proposed algorithms have far better performance than several baselines on continuous control. The network architecture of state-dependent adaptive temperature is clarified in Appendix E.

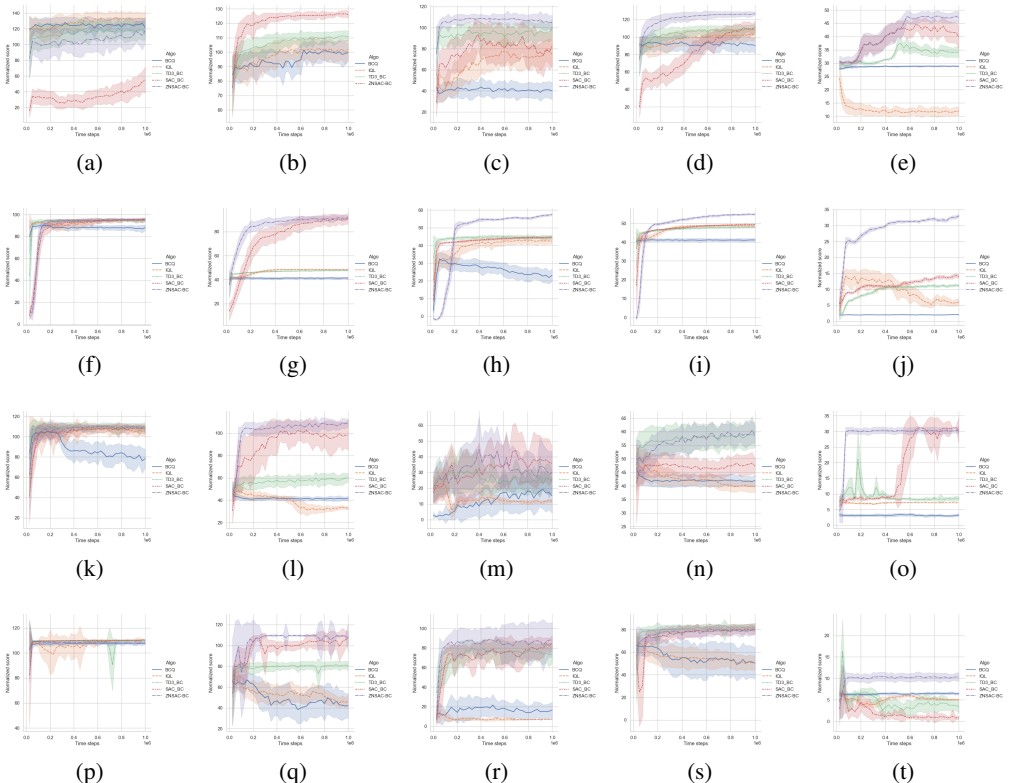

Figure 2: Normalized score versus time step in (a) Ant-expert; (b) Ant-medium-expert; (c) Ant-medium-replay; (d) Ant-medium; (e) Ant-random; (f) Halfcheetah-expert; (g) Halfcheetah-medium-expert; (h) Halfcheetah-medium-replay; (i) Halfcheetah-medium; (j) Halfcheetah-random; (k) Hopper-expert; (l) Hopper-medium-expert; (m) Hopper-medium-replay; (n) Hopper-medium; (o) Hopper-random; (p) Walker2d-expert; (q) Walker2d-medium-expert; (r) Walker2d-medium-replay; (s) Walker2d-medium; (t) Walker2d-random

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
