# OpenReview forum: "Z-score Normalized SAC Plus Behavioural Cloning for Offline Reinforcement Learning"
_ICLR.cc/2024/Conference — ICLR 2024 Conference Withdrawn Submission_

### Official Review · Reviewer_Q6ES · 2023-10-24

**Soundness:** 2 fair
**Presentation:** 2 fair
**Contribution:** 1 poor
**Rating:** 3
**Confidence:** 4

**Summary:**

Authors claim that replacing TD3 in TD3 + BC with SAC improves the performance and additionally propose modification for actor update which mitigate BC hyperparameter choice by normalizing Q value functions.

**Strengths:**

Proposed modification is motivated and potentially can be useful on practice. Author's experiments show that it is beneficial when applied to SAC+BC.

**Weaknesses:**

**Results presentation**

Table 1 contains only mean values without stds which must be included in RL tables.

Figure 2 is intended to present main author's results but it is really hard to read. Please report results in the table format in the main text and put training curves into appendix. I would also recommend to split this figure into multiple figures (e.g. by environment) and add dataset name as the title of each plot.

Please also add scores averaged over datasets.

**TD3 + BC scores**

Scores for TD3 + BC in the Table 1 are very strange. I've never seen it performing that bad. Could you please explain why this results are so different from other papers? SAC + BC scores seem to be close to the TD3 + BC scores I saw in other works which makes claim about SAC improvement questionable.

https://arxiv.org/abs/2110.06169
https://arxiv.org/abs/2206.04745
https://arxiv.org/abs/2210.07105
https://arxiv.org/abs/2305.09836

**Baselines choice**

IQL is the only strong base which authors compare against. For example, recent work ReBRAC (https://arxiv.org/abs/2305.09836), which is missed relevant work, modified TD3 + BC by applying some design choices from other algorithms resulting into much better performance while being TD3 + BC in fact.

There are also ensemble-based approaches like SAC-N/EDAC (https://arxiv.org/abs/2110.01548) or RORL (https://arxiv.org/abs/2206.02829) which are the strongest baselines for MuJoCo tasks. If would be fine if you used domains beside MuJoCo and compared only with ensemble-free baselines but only MuJoCo is used.


**Limited evaluation**

Authors tested their algorithm using only MuJoCo datasets which are not enough today in my opinion. For example, SAC-N/EDAC perform great in this domain but fails to learn anything on AntMaze. Please consider running your approach on more challenging D4RL domains like AntMaze, Adroit or Kitchen.


**Limited ablation study**

Modification for actor loss is tested only for SAC + BC.

What if we apply it to TD3 + BC? Can it boost ReBRAC's performance even further?

Will it benefit algorithm like IQL or CQL where we don't have the same BC term in actor's loss?

Is SAC + BC better in offline-online setup?

Does your loss modification benefit offline-to-online setup?

**Novelty**

From my point of view there is not enough novelty in the paper. Claim about SAC benefits is questionable because of TD3 + BC problems (see **TD3 + BC scores**). And loss modification is the only thing that can be seen as something useful. It could have been compensated with broader evaluation and ablations which are also very limited.

**Questions:**

All of the questions are taken from **Weaknesses**. I understand that not all of them can be answered during the short rebuttal phase but I kindly ask you to run more experiments in order to answer at least part of them.

* Could you please explain why TD3 + BC results are so different from other papers?

* What is the performance of your approach on AntMaze/Adroit/Kitchen D4RL tasks?

* What if we apply it to TD3 + BC? Can it boost ReBRAC's performance even further?

* Will it benefit algorithm like IQL or CQL where we don't have the same BC term in actor's loss?

* Is SAC + BC better in offline-online setup?

* Does your loss modification benefit offline-to-online setup?

---

### Official Review · Reviewer_mezX · 2023-10-31

**Soundness:** 1 poor
**Presentation:** 2 fair
**Contribution:** 1 poor
**Rating:** 3
**Confidence:** 5

**Summary:**

This paper aims to solve the inaccurate value estimation issue of the out-of-distribution actions. Compared with previous methods, this work gives a more accurate balance of the relative weight between RL and BC. The authors conduct experiments on standard benchmark and they find that BC term can be added to the policy update of SAC algorithm to get extensively better performance with proper weight adjustment and self-adaption.

**Strengths:**

1.	This paper is written well and easy to follow. The authors present a simple method to improve the current offline methods.
2.	This paper conducts corresponding experiments to preliminary verify their ideas.

**Weaknesses:**

1.	There is no detailed motivation for this paper. Why we need z-score normalization in offline RL?
2.	The experimental presentation is very poor. The label of figure 2 is very small. The paper does not have detailed ablation experiments.
3.	The paper does not rigorously verify the author's claims. The article is only 7 pages long. I believe that this paper is not ready for publication at this conference.

**Questions:**

1.	Why we need SAC+BC rather than TD3+BC? What is the difference between these two methods?
2.	Why we need z-score normalization? The paper claims that ‘Firstly, the distribution of z-score normalized modified Q-value is close to standard normal distribution, so that it can be balanced to suit the value of BC term’ and ‘Secondly, the approximated standard normal distribution can avoid the sensitivity to the absolute value of Q-value ‘. Could you please provide theoretical verification or experimental results to support this claim?
3.	The details of the experiment are not stated in the paper, such as Table 1.

---

### Official Review · Reviewer_jNsi · 2023-10-31

**Soundness:** 2 fair
**Presentation:** 2 fair
**Contribution:** 2 fair
**Rating:** 1
**Confidence:** 5

**Summary:**

The paper proposes ZNSAC-BC, a new offline reinforcement learning algorithm that combines soft actor-critic (SAC) with behavioral cloning. It applies z-score normalization to balance the RL and BC terms and make the Q-values more standard normal distributed, avoiding sensitivity to magnitudes. The trainable variance adapts policy gradients to mitigate extrapolation error. Experiments show ZNSAC-BC outperforms selected prior methods on some D4RL benchmarks, especially for lower expert datasets. The approach extends the TD3-BC framework with a simple but effective trick, providing an incremental improvement to the state-of-the-art. However, the gains seem modest and the paper lacks rigorous comparisons to prior methods. More benchmarking is needed to conclusively demonstrate superiority over existing algorithms. Overall, it makes an interesting contribution but needs stronger empirical validation.

**Strengths:**

Builds on simple and effective TD3-BC by changing the base RL algorithm to SAC. Makes intuitive sense.

Z-score normalization of Q-values is a nice trick to balance terms and adapt gradients.

Overall a straightforward extension of prior work with clear improvements.

**Weaknesses:**

While better than TD3-BC, the gains seem somewhat incremental, not a dramatic breakthrough.

Lack of comparisons with SOTA methods

The experimental evaluation is very limited (only "easy" MuJoCo tasks) and  empirical results are underwhelming when compared to similar approaches

More analysis and intuition explaining why the z-score normalization helps would strengthen the approach.

The writing could be tightened up and made easier to follow in some parts.

**Questions:**

Report numerical results for ZNSAC-BC and baseline methods on the D4RL benchmarks.

The limited reported results look fine when compared to your chosen algorithms, but fall short of those in other recent papers.  This is particularly the case for medium-replay and medium datasets where it looks like you do well when compared to TD3-BC, SAC-BC and IQL, but are inferior to other algorithms particularly for hopper/walker2d.

Provide tables aggregating final performance across multiple runs with standard deviations, to show statistical significance of the improvements, and add quantitatively compare performance to the published state-of-the-art results on these benchmarks from prior papers, to put the gains in context.

**Details Of Ethics Concerns:**

No concerns

---

### Official Review · Reviewer_whc9 · 2023-10-31

**Soundness:** 3 good
**Presentation:** 1 poor
**Contribution:** 2 fair
**Rating:** 3
**Confidence:** 4

**Summary:**

The paper proposes a new algorithm called z-score normalized SAC-BC (ZNSAC-BC), tackling the overestimation issue in offline reinforcement learning. ZNSAC-BC uses a behavioral cloning (BC) term to constrain the learned policy to not bootstrap from out-of-distribution actions, thus mitigating the overestimation. The paper also introduces a normalization term of the action value to balance the value estimation and the BC term in the policy improvement step.

**Strengths:**

- The paper aims to mitigate the overestimation issue in offline reinforcement learning. The problem it focuses on is meaningful and useful.

- The method proposed is novel and empirically showed a reasonable performance.

**Weaknesses:**

- The paper is inconsistent. It seems like the main algorithm introduced in the paper is ZNSAC-BC, which is an offline learning algorithm for mitigating the overestimation issue caused by out-of-distribution action. However, the conclusion section talks about a totally different algorithm called AAAC, which mainly focuses on a state-dependent adaptive temperature for stabilizing training. Meanwhile, the abstract indicates the proposed method is called \emph{SAC-BC}. ZNSAC-BC performed better than SAC-BC empirically, according to the experiment section, so I think it might be better to claim the main contribution is ZNSAC-BC instead of SAC-BC in the abstract.

- The labels in Figure 2 are small and hard to read. The color in subplot b is inconsistent with other subplots, and the result for SAC-BC is missing.

- Detailed experiment settings are not provided. The missing information reduces the reproducibility of the experiment.

A Small Thing:

- For the left quotation mark, you may use `` instead of “.

**Questions:**

- ZNSAC-BC introduces a new parameter, $\rho$, which seems manually selected and remains constant during training. The paper mentions $\rho$ is set to 2.5 in the experiment, without explaining how this number was selected. It could be better if the paper includes either an empirical study regarding how sensitive the method is to $\rho$, or a guide on how to select this parameter.

- The algorithm maintains target networks for the actor and the critic. Both target networks are used when calculating the bootstrapping target for the critic. Usually, only maintaining a target network for the critic is enough for learning. That is, the next action is sampled from the learning actor network instead of the target actor network. I would like to ask if the authors investigate the advantage of sampling the next action from the target actor network, comparing to sampling it from the learning actor network. Moreover, in the experiment section, did all baselines use the same setting, i.e., maintaining target networks for both actor and critic networks, and sampling action from the target actor network when calculating the bootstrapping target?